# BONSAI: Depth-Constrained Decision Tree Induction via Approximation Optimization

## Abstract

We propose BONSAI (Branch Optimization for Numerical Splits for Accurate and Interpretable models), a novel decision tree algorithm that optimizes split decisions over numerical attributes under a depth limit. In contrast to conventional greedy methods such as CART and C4.5, which construct trees via locally top-down splits, BONSAI formulates tree induction as an approximation optimization problem based on a new slot-node structure that can deal with both categorical and numerical features. To improve the performance and interpretability, BONSAI combines a compact optimization formulation with efficient search-space reduction techniques, thereby avoiding the combinatorial overhead of traditional mixed-integer programming (MIP) approaches. Empirical evaluations on benchmark datasets show that BONSAI achieves superior predictive accuracy, model compactness, and interpretability compared to both greedy and existing optimization-based methods.

## 1 Introduction

Decision trees (DTs) are interpretable predictive models structured as hierarchical IF-THEN rules, contrasting with black-box models like neural networks. Their inductive processes have gained attention for decision-making due to their alignment with human reasoning Chen & Rudin (2018). Classical top-down algorithms such as CART(Classification And Regression Tree) Breiman et al. (1984) and C4.5 Quinlan (1993) recursively split nodes based on impurity measures (e.g., Gini index, information gain), offering efficiency and scalability. However, these greedy methods optimize splits locally and often yield suboptimal trees by ignoring global interactions between splits Bertsimas & Dunn (2017).

Formulating tree construction as a joint optimization problem can address these limitations by enabling jointly optimized split decisions. Combined with our candidate-generation scheme (in Section 4.2), the resulting MIP is solved to optimality over the reduced candidate set—a setting we refer to as approximation optimization. Approaches based on mixed-integer programming (MIP), dynamic programming (DP), and constraint programming (CP) have been proposed, particularly for binary trees as shown in Table 1. Despite their principled nature, these methods face severe scalability challenges. Even shallow optimal trees are difficult to construct, as the problem is NP-hard Laurent & Rivest (1976), and the search space grows exponentially with depth. For instance, Verwer & Zhang (2019) show that a MIP solver fails to find an optimal depth-2 tree within 10 minutes on small datasets.

In addition to scalability, binary branching limits the expressive capacity of shallow trees, exacerbating the trade-off between accuracy and interpretability. Constraining tree depth improves interpretability but reduces modeling flexibility. To address this, we propose a novel tree learning algorithm that jointly optimizes multiple splits over numerical features within a fixed depth. Our method leverages efficient search and mathematical modeling to enhance predictive accuracy while maintaining interpretability.

## 2 Related Work

Table 1 highlights the breadth of recent work on learning *optimization-based* DTs. Researchers have proposed formulations based on MIP, DP, and CP, each optimizing different objectives—most

Table 1: Comparison of optimization-based DT algorithms with respect to their mathematical formulation, supported feature types, and key algorithmic characteristics. Numerical and categorical features are denoted by $N$ and $C$, respectively. Categorical variables encoded via one-hot binarization are indicated as $C^{\mathrm{O}}$, whereas $C^{\mathrm{i}}$ designates itemset-based binarization.

| Reference | Model | Objective | Features | Splits on $N$ | Structure |
|---|---|---|---|---|---|
| Bertsimas & Dunn (2017) | MIP | Accuracy, Complexity | $N$ | Binary | Imbalanced |
| Verwer & Zhang (2019) | MIP | Accuracy | $N$ | Binary | Balanced |
| Aglin et al. (2020) | MIP | Accuracy | $N, C^i$ | Binary | Imbalanced |
| Demirović et al. (2022) | DP | Accuracy | $N, C^o$ | Binary | Imbalanced |
| Lin et al. (2020) | DP | Accuracy, $F_1$ | $N, C^o$ | Binary | Imbalanced |
| Aglin et al. (2020) | CP | Accuracy | $N, C^i$ | Binary | Imbalanced |
| Günlük et al. (2021) | MIP | Accuracy | $N, C^o$ | Binary | Imbalanced |
| Mazumder et al. (2022) | MIP | Accuracy | $N, C^o$ | Binary | Balanced |
| Aghaei et al. (2024) | MIP | Accuracy | $N, C^o$ | Binary | Balanced |
| van der Linden et al. (2023) | DP | Nonlinear objectives (e.g., $F_1$) | $N, C^o$ | Binary | Imbalanced |
| **Our approach** | MIP | **Accuracy, Complexity** | $N, C$ | **Multiple** | **Imbalanced** |

commonly overall accuracy, but sometimes combinations of accuracy with complexity or $F_1$ score. Feature support likewise varies: early methods handle mainly numerical features ($N$), whereas more recent approaches incorporate categorical variables via binarization such as one-hot encoding ($C^{\mathrm{o}}$) or itemset encodings ($C^{\mathrm{i}}$). Despite this diversity, nearly all existing models restrict numerical attributes to binary splits and often enforce balanced tree structures, which can limit performance and interpretability under depth constraints.

Optimization models that binarize categorical variables—whether through one-hot or itemset encodings—often undermine both interpretability and predictive performance, especially when tree depth is constrained. One-hot encoding splinters a single categorical attribute into many binary indicators, forcing a binary DT to string together multiple sequential splits to replicate what a single multi-way split would capture Dorogush et al. (2018). With a limited depth budget, the tree must either accept missed interactions or devote several levels to separating categories, quickly exhausting its depth. Itemset encodings exacerbate the problem: by adding a binary feature for every category combination, they inflate the feature space and fragment the data so severely that a shallow tree cannot feasibly explore the combinatorial explosion of candidate splits, leading to overfitting or overlooked patterns. Echoing these concerns, Aghaei et al. (2019)) report that DTs trained on one-hot inputs are markedly less interpretable, whereas models that handle categorical variables directly produce more compact, flexible, and human-readable trees.

To overcome these limitations, we introduce an optimization-based DT framework that natively handles both numerical and categorical features and permits multi-way splits on numerical attributes, thereby boosting predictive performance while preserving interpretability.

## 3 PROBLEM DEFINITION

Consider a supervised learning problem with $n$ observations $\mathcal{D} = \{(x_i, y_i)\}_{i \in \mathcal{I}}$, each with $f \in \mathcal{F}$ features $x_i \in \mathcal{X} \subseteq \mathbb{R}^f$ and response $y_i \in \mathcal{Y}$. A DT recursively partitions the feature space $\mathcal{X}$ into a number of hierarchical, disjoint regions, and makes a prediction for each region. For a DT $T$ and feature vector $x$, let $T(x) \in \mathcal{Y}$ denote the corresponding prediction. For classification problems with $K$ classes, $\mathcal{Y} = \mathcal{K} = \{1, \dots, K\}$ and $\ell_{01}$ is the 0-1 (or mis-classification) loss where $\ell_{01}(y, \hat{y}) = 0$ if $y = \hat{y}$; otherwise, 1. Given a loss function $\ell_{01}(\cdot, \cdot)$ on $\mathcal{Y} \times \mathcal{Y}$ and a family of $\mathcal{T}$ of DTs, finding an optimal DT is represented as the following optimization problem:

$$\min_{T \in \mathcal{T}} \sum_{i=1}^{n} \ell_{01}(y_i, T(x_i)) \tag{1}$$

In addition to minimizing classification error, maintaining the interpretability of DTs typically involves two regularization strategies: (1) limiting the tree depth and (2) reducing the number of

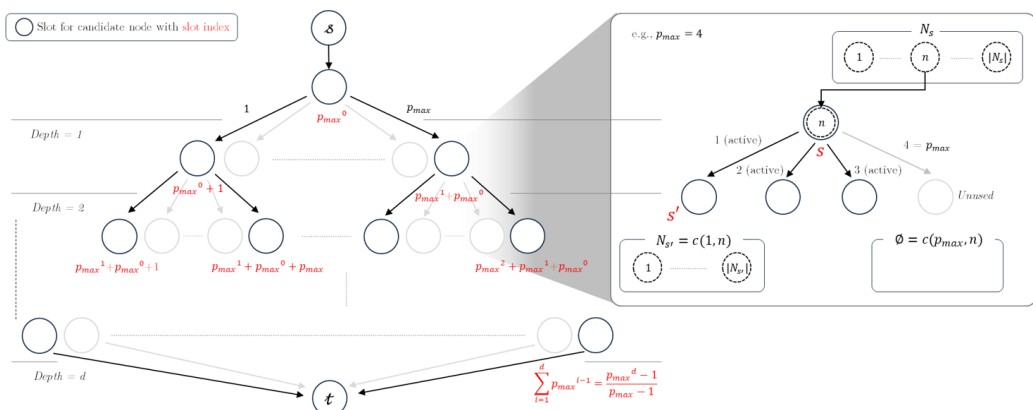

Figure 1: Slot-node structure, candidate nodes for each slot, and their relationships.

decision rules. Interpretable models are generally expected to be compact, making depth control a key factor; for example, a tree of depth 5 is substantially more interpretable than one of depth 50 (Good et al. 2023). To quantify complexity in terms of decision rules, Moshkovitz et al. (2021) introduce the interpretation complexity (IC), defined as the number of distinct feature-threshold pairs used in the tree. To jointly minimize classification error and IC, subject to a maximum depth constraint $N_{\max}$, we define the following lexicographic optimization objective:

$$\operatorname*{lex\,min}_{T \in \mathcal{T}} \left( \sum_{i=1}^{n} \ell_{01}(y_i, T(x_i)),\ \mathrm{IC}(T) \right) \quad \text{subject to} \quad depth(T) \leq N_{\max} \tag{2}$$

where $depth(T)$ denotes the maximum depth of $T$. We consider the general setting where each input $x_i$ comprises both numerical and categorical features, indexed by $f \in F$ with $F = F_{\text{cat}} \cup F_{\text{num}}$. For a numerical feature $f \in F_{\text{num}}$, the set of values $\{x_{(i,f)}\}_{i \in I}$ can contain at most $|I|$ distinct elements. Even for balanced binary DTs, computing the optimal tree structure requires $\mathcal{O}(n^d p^d)$ operations, where $n$ denotes the number of unique feature values, $p$ the number of features, and $d$ the tree depth Mazumder et al. (2022). Thus, in the following section, we present a novel optimization framework that integrates search space reduction techniques to efficiently identify high-performing trees within a practical computational budget. We do not attempt to solve (1)–(2) over the entire search space. Instead, we restrict the space to trees whose internal nodes and thresholds are drawn from the candidate sets generated by Algorithms 1–2, and we solve the resulting MIP to optimality on this pruned space. We refer to this setting as approximation optimization.

## 4 BONSAI: DEPTH-CONSTRAINED DECISION TREE INDUCTION VIA APPROXIMATION OPTIMIZATION

In this section, we propose a novel optimization framework, **BONSAI** (*Branch Optimization for Numerical Splits for Accurate and Interpretable models*), which identifies high-performing DTs by reducing the candidate node search space and employing a MIP formulation with lower computational complexity than existing MIP-based methods. Throughout, 'approximation optimization' denotes exact MIP optimization on a search space compressed by our slot–node structure and the two-step candidate selection ($\rho$ and $\tau$), rather than global optimization over all possible trees. Table 2 summarizes the notations of indices, datasets, functions, sets, parameters, and decision variables used in BONSAI.

### 4.1 SLOT-NODE STRUCTURE

In this paper, we propose a slot-based framework wherein each slot represents a set of feasible positions for nodes and maintains a corresponding set of candidate nodes. The slot-based structure offers two main advantages: (1) it enables the construction of imbalanced trees by defining candidate nodes per slot, and (2) it facilitates search space reduction by selectively narrowing the dataset. As illustrated in Figure 1, the complete slot structure is predefined prior to evaluating candidate nodes

Table 2: Summary of notation used in BONSAI.

| Category | Notation | Description |
|----------|----------|-------------|
| *Indices* | $d$ | Depth (0 means a root of tree) |
| | $p$ | Branch position ($p \in \mathcal{P} := \{1, \ldots, p_{\max}\}$) |
| | $s$ | Slot ($s \in \mathcal{S} := \{1, \ldots, s_{\max}\}$) |
| | $f$ | Feature $f \in F$, $F = F_{\text{cat}} \cup F_{\text{num}}$ |
| | $k$ | Class $k \in K$ |
| | $n$ | Candidate node $n \in N_s$ where $s$ represents a given slot |
| | $i$ | Datapoint $i \in I$ |
| *Datasets* | $D$ | Initial dataset |
| | $D_n$ | Set of instances that reach node $n$ |
| | $I_n$ | Set of datapoints in $D_n$ |
| | $z_n^i$ | $z_n^i = 1$ if node $n$ has a directed flow to terminal node $t$ and datapoint $i$ is correctly classified; otherwise $z_n^i = 0$ |
| *Functions* | $u(f, D)$ | Number of distinct values in $f$ at $D$ |
| | $p(n)$ | Return set of parent nodes for node $n$ |
| | $\text{pos}(\cdot)$ | Return a position of the node in $p(n)$ or slot $s$ |
| | $\text{up}(s)$ | Return parent slot of the given slot $s$ |
| | $a(p, n)$ | A binary matrix-valued function indicating whether an active child node exists at position $p$ for node $n$: $a(p, n) = 1$ if at least one child exists at position $p$, and $a(p, n) = 0$ otherwise. |
| | $c(p, n)$ | Set of all child nodes at position $p$ If $a(p, n) = 0$, then $c(p, n) = 0$; otherwise $|c(p, n)| > 0$ |
| | $\text{depth}(s)$ | Return the depth of slot $s$ |
| *Sets* | $N_s$ | Set of nodes in slot $s$ |
| | $\Theta_f$ | Set of thresholds for feature $f$ |
| | $\mathcal{T}$ | Set of top branching nodes |
| | $\mathcal{B}$ | Set of branching nodes |
| | $\mathcal{L}$ | Set of leaf nodes |
| | $\mathcal{N}$ | All nodes across all slots, $\mathcal{N} = \sum_{s \in \{1, \ldots, s_{\max}\}} N_s$ |
| *Parameters* | $\mathfrak{s}$ | Starting node |
| | $\mathfrak{t}$ | Terminal node |
| | $d_{\max}$ | Maximum tree depth |
| | $b_{\max}$ | Maximum number of branches |
| | $N_{\max}$ | Maximum number of decision nodes |
| | $\rho$ | Sampling ratio |
| | $\tau$ | Distance tolerance |
| *Decision variable* | $b_n$ | $b_n = 1$ if node $n$ is selected; else $b_n = 0$ |

for each slot. Given the dataset and parameters—specifically, the maximum number of branches and the number of categories—the maximum number of branches per slot, denoted as $p_{\max}$ is computed as $p_{\max} := \max\left(b_{\max}, \max_{f \in \mathcal{F}_{\text{cat}}} u(f, \mathcal{D}_n)\right)$, where $b_{\max}$ is the maximum allowable number of branches.

The maximum number of slots, denoted as $s_{\max}$ is given by $s_{\max} := \sum_{i=1}^{d} p_{\max}^{i-1} = \frac{p_{\max}^d - 1}{p_{\max} - 1}$, where $d$ is the depth of the tree. After defining the slot structure, the search procedure described in Algorithm 1 begins by identifying all candidate nodes for each slot, updating the filtered dataset at each specific node, denoted as $D_n$. Each node is associated with a parent node and one or more child nodes, where the child nodes correspond to different branching strategies for continuous features. Given the data at a specific node $n$, the following splitting strategy determines candidate thresholds for multi-way splits, which in turn influence the number of child nodes and the overall computational complexity.

**Split Strategy** Given the data at a specific node $(n)$ $\mathcal{D}_n = \{(x_i, y_i)\}_{i \in I_n}$ with $I_n \subseteq I$, and a numerical feature $f \in F_{\text{num}}$, let $u(f, \mathcal{D}_n)$ denote the number of distinct values in $\{x_{(i,f)}\}_{i \in I_n}$. Let $w_1^f < w_2^f < \cdots < w_{u(f, \mathcal{D}_n)}^f$ represent these distinct values in $\{x_{(i,f)}\}_{i \in I_n}$ and define $w_0^f = -\infty$ and $w_{u(f, \mathcal{D}_n)+1}^f = \infty$. For any integer $j$ with $0 \leq j \leq u(f, \mathcal{D}_n)$, define $\tilde{w}_j^f := \frac{w_j^f + w_{j+1}^f}{2}$ as the representative threshold. Define $\mathcal{C}_{\mathcal{D}_n}^f$ as the set of ordered set of thresholds for the numerical feature

$f$ at node $n$, considering a maximum number of branches $b_{\max}$, as follows:

$$\mathcal{C}_{\mathcal{D}_n}^f := \left\{ C = \left(c^{(1)}, \ldots, c^{(|C|)}\right) \middle| C \in \mathcal{P}\left(\left\{\tilde{w}_j^f\right\}_{1 \leq j \leq u(f, \mathcal{D}_n)}\right), \ |C| < b_{\max} - 1, \ c^{(1)} < \cdots < c^{(|C|)} \right\} \tag{3}$$

where $\mathcal{P}(\cdot)$ denotes the power set. For categorical features, splitting thresholds are unnecessary; instead, nodes partition the data by enumerating all categories, resulting in one branch per category to leverage the advantages of multiway splits under a depth limitation. If certain categories or branches are absent from $D_n$, branches corresponding to absent branches are marked as no data (ND) and assigned an arbitrary label. Under this setting, in the worst-case scenario where all features are continuous (i.e., $\mathcal{F} = \mathcal{F}_{\text{num}}$) and the dataset cannot be partitioned at any split, the cost of exhaustive search is $\mathcal{O}\left(\left(|\mathcal{F}| \cdot 2^{\max_{f \in \mathcal{F}} u(f, \mathcal{D})}\right)^{s_{\max}}\right)$ as detailed in Appendix A.1. To efficiently identify a high-performing tree within a reasonable time, we propose search strategies for candidate node selection, along with a compressed mathematical formulation in the following section.

**Algorithm 1**    SEARCHING FOR CANDIDATE NODES$(D, b_{\max}, d_{\max}, \rho, \eta)$
**Require:** $D, b_{\max}, d_{\max}, \rho, \tau$
**Ensure:** $\mathcal{N}, z_n^i, \mathcal{T}, \mathcal{B}, \mathcal{L}, a(\cdot)$
1: Initialize $\mathcal{N} \leftarrow \{N_s\}_{s=1}^{s_{\max}}$
2: $\mathcal{D} := \{D_1, \ldots, D_n\}$, where $D_i = \{D_n^{(1)}, \ldots, D_n^{(p)}\}$
3: Initialize $\mathcal{T}, \mathcal{B}, \mathcal{L}, a(\cdot), z_n^i$
4: **for** $s \in \mathcal{S}$ **do**
5:     **for** $f \in F$ **do**
6:         **if** $s = 1$ **then**
7:             SEARCH_NODE$(s, f, D)$
8:         **else**
9:             **for all** $n \in \{n' \in N_{\text{up}(s)} \mid a(\text{pos}(s), n') = 1\}$ **do**
10:                SEARCH_NODE$(s, f, D_n^{(\text{pos}(s))})$
11:            **end for**
12:        **end if**
13:    **end for**
14: **end for**
15: **return** $\mathcal{N}, z_n^i, \mathcal{T}, \mathcal{B}, \mathcal{L}, a(\cdot)$

## 4.2    SEARCHING FOR CANDIDATE NODES

To reduce the search space while minimizing performance degradation, we propose a two-step procedure for searching candidate nodes: (1) identifying nodes with minimal threshold distances and (2) performing random sampling without replacement.

**Minimal threshold distances:** This approach is motivated by the assumption that closely spaced thresholds may lead to overly complex trees with poor generalization. Empirical evidence from Amro et al. (2021) supports this notion, showing that overfitted DTs often exhibit jagged decision boundaries that tightly conform to borderline or noisy instances. Enforcing a minimum distance between split thresholds can improve DT efficiency and effectiveness. By avoiding nearly identical splits, the tree is less likely to chase noise (overfitting) and tends to be more interpretable (since splits occur at more meaningful intervals). We define $\tau$ as user-defined distance tolerance to appropriately space thresholds for numerical features based on the number of branches, and extract a subset of candidate nodes whose pairwise threshold distances satisfy this constraint, as follows:

$$\mathcal{C}_{D_n}'^f := \left\{ C \in \mathcal{P}\left(\{\tilde{w}_t^f\}_{1 \leq t \leq u(f, D_n)}\right) \middle| |g - h| \geq \tau \cdot \frac{\max_j \tilde{w}_j^f - \min_j \tilde{w}_j^f}{b_{\max}} \quad \forall g, h \in C \right\} \tag{4}$$

**Threshold Sampling:** This approach is inspired by Extremely Randomized Trees (Extra-Trees) proposed by Geurts et al. (2006), which reduce the computational cost of tree construction by sampling split thresholds uniformly at random, avoiding exhaustive threshold searches. Similarly, we sample thresholds without replacement, using a sampling ratio $\rho$ to determine the number of candidate thresholds. Specifically, the candidate set size is proportional to $C_{D_n}'^f$ the size of the previously extracted subset.

$$C_{D_n}''^f \sim \text{Unif}\left\{ C : C \subseteq C_{D_n}'^f, \ |C| = \left\lceil \rho \cdot |C_{D_n}'^f| \right\rceil \right\} \tag{5}$$

Based on the two-step procedure, the overall algorithm for identifying candidate nodes and retrieving associated information—such as the partial dataset $D_n$ and classification errors $z_n^i$ for each node—is summarized in the following pseudocode.

**Algorithm 2** FUNCTION SEARCH_NODE$(s, f, D)$

1: **if** $(s + 1 = d_{\max})$ or depth$(s) = d_{\max}$ or all labels in $D$ are equal or $D = \varnothing$ **then**

2:     Create a leaf node: $\mathcal{L} \leftarrow \mathcal{L} \cup \{n\}$, update $z_n^i = \mathbb{1}[i \in I_n] \cdot \mathbb{1}\left[y_i = \arg\max_{k \in \mathcal{K}} \sum_{j \in I_n} \mathbb{1}[y_j = k]\right]$

3:     **for** $p \in \mathcal{P}$ **do**

4:         $a(p, n) \leftarrow 0$

5:     **end for**

6: **else**

7:     $\mathcal{C}'^f_{D_n} := \left\{ C \in \mathcal{P}\left(\{\tilde{w}_t^f\}_{1 \le t \le u(f, D_n)}\right) \;\middle|\; |g - h| \ge \tau \cdot \frac{\max_j \tilde{w}_j^f - \min_j \tilde{w}_j^f}{b_{\max}} \quad \forall g, h \in C \right\}$

8:     $C''^f_D \leftarrow \text{UNIF}\left\{ C \subseteq C^f_D \;\middle|\; |C| = \left\lfloor \rho \cdot |C^f_D| \right\rfloor \right\}$

9:     **for** $C \in C''^f_D$ **do**

10:         Create a node $n$ and add to $\mathcal{N}_s$

11:         **if** $s = 1$ **then**

12:             $\mathcal{T} \leftarrow \mathcal{T} \cup \{n\}$

13:         **else**

14:             $\mathcal{B} \leftarrow \mathcal{B} \cup \{n\}$

15:         **end if**

16:         **if** $f \in F_{\text{num}}$ **then**

17:             $\Theta_f \leftarrow \{(-\infty, c^{(1)}], [c^{(1)}, c^{(2)}), \dots, [c^{(|C|)}, \infty)\}$

18:         **else**

19:             $\Theta_f \leftarrow \{u(f, D)\}$

20:         **end if**

21:         **for** $p \in \mathcal{P}$ **do**

22:             **if** $p \le |\mathcal{C}|$ **then**

23:                 $a(p, n) \leftarrow 1$

24:                 $D_n^{(p)} \leftarrow \{(x_i, y_i) \in D \mid x_{i,f} \in \Theta_f^{(p)}\}$

25:             **else**

26:                 $a(p, n) \leftarrow 0, D_n^{(p)} \leftarrow \varnothing$

27:             **end if**

28:         **end for**

29:         $D \leftarrow D \cup \bigcup_{p=1}^{p_{\max}} D_n^{(p)}$

30:     **end for**

31: **end if**

## 4.3 MIP FORMULATION

Based on the indices, sets, parameters, and decision variables defined in Table 1, the proposed mathematical formulation is given as follows:

$$\text{lex min} \quad (f_1, f_2) \tag{6}$$

$$f_1 = |I| - \sum_{n \in \mathcal{L}} b_n \left( \sum_{i \in I} z_n^i \right) \tag{7}$$

$$f_2 = \sum_{n \in \mathcal{N}} b_n \tag{8}$$

$$\text{subject to} \quad \sum_{n \in \mathcal{T}} b_n = 1 \tag{9}$$

$$a(p, n) \cdot b_n = \sum_{n'' \in \{n' \in c(p,n) \mid \text{pos}(n') = p\}} b_{n''} \qquad \forall p \in \mathcal{P}, \, \forall n \in \mathcal{T} \cup \mathcal{B} \tag{10}$$

$$\sum_{p \in \mathcal{P}} a(p, n) \cdot b_n = \sum_{n' \in c(p,n)} b_{n'} \qquad \forall p \in \mathcal{P}, \forall n \in \mathcal{B} \tag{11}$$

$$\sum_{n \in \mathcal{N}} b_n \le N_{\max} \tag{12}$$

$$b_n \in \{0, 1\}, \quad \forall n \in \mathcal{N} \tag{13}$$

In this formulation, the objective function (6) performs a lexicographic minimization of (7) classification errors ($f_1$) and (8) the number of decisions in the tree ($f_2$), where the latter corresponds to the interpretation complexity (IC), defined as the number of feature-threshold pairs Moshkovitz et al. (2021) . Constraint (9) enforces that exactly one node is selected at the root level. Constraint (10) ensures that a non-leaf child node can be activated only if its parent node has an active branch at the corresponding branching position. Constraint (11) enforces consistency in outflow, requiring that if a parent node is selected, the number of active child nodes matches the number of branches for that node. Constraint (12) restricts the total number of selected nodes to not exceed a specified IC limitation. This MIP is solved to optimality on the reduced candidate space induced by Algorithms 1–2; we emphasize that this constitutes approximation optimization with respect to the unrestricted problem in Section 3.

## 5 EXPERIMENTAL RESULTS

In this section, we evaluate the performance of BONSAI across a variety of datasets to assess its effectiveness in learning underlying patterns using a single DT. Our experimental setup follows prior work on optimization-based DTs, which compares the performance of single trees or ensembles of trees to highlight differences in predictive accuracy and interpretability. To ensure a fair comparison among single DT models, we constrain the maximum depth to 3, aligning with prior work that emphasizes interpretability Souza et al. (2022). The experiments were conducted on a system equipped with an Intel i9-14900 CPU, 64 GB of RAM, and an NVIDIA RTX 4080 Ti GPU.

We compare our method against both greedy and optimization-based DT algorithms. For greedy methods, we consider the most widely used CART Breiman et al. (1984) and C4.5 algorithms Quinlan (1993), which rely on impurity measures such as Gini index and Information Gain, respectively. For optimization-based approaches, we evaluate Quant-BnB Mazumder et al. (2022) and Fast Interpretable Greedy-tree Sums (FIGS) Tan et al. (2025). Quant-BnB is a state-of-the-art method that learns balanced binary DTs via a MIP formulation. We compare its predictive performance against BONSAI, which induces imbalanced trees with multi-way splits. FIGS, in contrast, is a recent ensemble method that controls model complexity through the number of rules rather than tree depth. We evaluate its predictive accuracy and interpretability in comparison to depth-constrained single-tree models. For FIGS, which does not expose a depth parameter but instead limits the number of rules, we set the rule cap to 10 to maintain a comparable level of model complexity. Other algorithms that require additional binarization of numerical attributes were excluded, as this preprocessing step can significantly affect performance and incur substantial computational overhead. The implementations of CHAID, Quant-BnB and FIGS used in our experiments are publicly available at `CHAID` package, `https://github.com/mengxianglgal/Quant-BnB`, and `https://github.com/csinva/imodels`, respectively. For FIGS, we employ the `OneVsRestClassifier` wrapper to support multiclass classification. The train- and test-splits used in our experiments, is publicly available at `https://github.com/bonsai-tree-model/bonsai`.

### 5.1 DATASETS

We evaluate our method on seventeen public classification datasets, vary widely in size (101–3 810 instances), feature type (purely categorical to purely numerical), class count (2–10) and imbalance ratio (1.00–18.62) as shown in Table 4 in the Appendix A.2. The heterogeneity of this test bed is intended to stress-test a depth-constrained tree learner across the principal challenges faced by interpretable models: mixed data types, skewed class distributions and small-sample regimes. For DT algorithms that do not natively support categorical features, categorical attributes are one-hot encoded. Each dataset is randomly split into 70% training and 30% test sets.

### 5.2 DISCUSSION OF EXPERIMENTAL RESULTS

**Overall performance** The quantitative results in Table 6 show that BONSAI attains the best test accuracy on 11 / 17 benchmarks (65 %), beating every baseline by a comfortable margin in most cases. Average ranks on the test split are 1.53 (BONSAI), 2.82 (FIGS), 3.00 (Quant-BnB), 3.41 (C4.5) and 4.24 (CART). Particularly large gains are observed on HEPATITIS (+13 pp over the runner-up) and ILPD (+3 pp), both of which contain mixed features and moderate imbalance, indicating that

Table 3: Training and test accuracy (%) across datasets and algorithms. An asterisk (*) denotes an ensemble model comprising multiple trees. Bolded values indicate the highest test accuracy for each dataset.

| Dataset | BONSAI | | CHAID | | C4.5 | | CART | | Quant-BnB | | FIGS* | |
|---|---|---|---|---|---|---|---|---|---|---|---|---|
| | Train | Test | Train | Test | Train | Test | Train | Test | Train | Test | Train | Test |
| Iris | 100.00 | **98.10** | 33.33 | 33.33 | 97.14 | 88.89 | 98.10 | 97.78 | 100.00 | 91.11 | 100.00 | 91.11 |
| Rice | 93.29 | **92.74** | 57.22 | 57.22 | 93.48 | 92.04 | 93.33 | 92.21 | 94.04 | 91.86 | 93.48 | 91.16 |
| Penguin | 98.71 | **100.00** | 43.78 | 43.56 | 96.14 | 96.04 | 98.71 | 98.02 | 100.00 | 97.03 | 100.00 | 97.03 |
| Blood | 81.07 | **81.33** | 76.29 | 76.00 | 77.63 | 76.89 | 78.78 | 78.67 | 80.69 | 80.44 | 80.88 | 79.11 |
| ILPD | 76.71 | **74.86** | 71.32 | 71.43 | 72.55 | 72.00 | 77.94 | 68.00 | 79.66 | 67.43 | 79.75 | 70.69 |
| Zoo | 95.71 | 93.55 | 40.00 | 41.94 | 82.86 | 83.87 | 82.86 | 83.87 | 94.29 | 93.55 | 100.00 | **96.77** |
| Wholesale | 76.62 | **73.48** | 71.75 | 71.97 | 72.40 | 71.97 | 72.73 | 66.67 | 77.27 | 68.18 | 82.14 | 62.12 |
| Hepatitis | 100.00 | **95.45** | 82.76 | 86.36 | 89.66 | 81.82 | 98.28 | 72.73 | 100.00 | 86.36 | 100.00 | 81.82 |
| Wine | 98.39 | **100.00** | 40.32 | 38.89 | 99.19 | 94.44 | 99.19 | 98.15 | 100.00 | 92.59 | 100.00 | 87.04 |
| Contraceptive | 58.20 | 56.79 | 42.68 | 42.76 | 50.44 | 48.42 | 52.67 | 54.52 | 57.90 | 56.79 | 60.04 | **58.60** |
| Diabetes | 79.52 | **76.19** | 65.18 | 64.94 | 77.47 | 73.59 | 78.96 | 72.73 | 81.75 | 71.43 | 80.82 | 74.89 |
| Tic-tac-toe | 81.49 | 65.28 | 65.37 | 65.28 | 78.06 | 71.88 | 76.87 | 71.88 | 78.81 | **73.96** | 74.93 | 71.53 |
| Car Evaluation | 84.37 | 70.71 | 70.06 | 69.94 | 83.87 | 79.00 | 81.97 | 77.26 | 82.22 | 78.61 | 93.30 | **92.29** |
| Image | 84.32 | 84.68 | 14.31 | 14.31 | 57.02 | 57.00 | 57.06 | 56.94 | 88.85 | 85.55 | 96.60 | **94.52** |
| Banknote | 94.38 | 95.63 | 55.52 | 55.58 | 92.60 | 89.81 | 94.58 | 92.48 | 98.23 | **98.54** | 98.23 | 97.29 |
| Statlog | 77.00 | **75.00** | 70.00 | 70.00 | 72.86 | 72.67 | 75.00 | 72.33 | 77.57 | 69.00 | 78.00 | 72.33 |
| Credit | 88.91 | **83.42** | 55.49 | 55.56 | 88.04 | 82.38 | 88.91 | 79.27 | 90.65 | 80.31 | 92.17 | 80.83 |
| Average | 86.39 | **83.37** | 56.20 | 56.42 | 81.26 | 78.39 | 82.70 | 78.15 | 87.17 | 81.34 | 88.84 | 82.30 |

the slot-node approximation optimization makes better use of heterogeneous predictors than greedy baselines. Across a diverse set of interpretable-scale benchmarks, BONSAI consistently delivers the best or second-best accuracy while maintaining a small generalisation gap. The results corroborate our hypothesis that *approximation optimization under explicit complexity budgets* yields DTs that are both compact and highly predictive, outperforming greedy (C4.5, CART, and CHAID) and existing optimization-based (FIGS, Quant-BnB) methods in the majority of cases.

**Generalization and trade-offs**  Although FIGS and Quant-BnB often drive the training error to zero, this incurs substantial over-fitting: their average train–test gaps are 6.5 pp and 5.8 pp, respectively, more than double BONSAI's 3.0 pp. The tighter gap suggests that BONSAI's joint optimization and built-in pruning (via the depth and IC budgets) effectively regularize the tree without sacrificing accuracy. Even though BONSAI often reaches the highest training accuracy, the combination of (i) an explicit depth budget and (ii) candidate-node pruning techniques prevents severe overfitting. Greedy methods such as C4.5 and CART mitigate overfitting by enforcing shallow depth constraints, but this restraint often reduces accuracy—especially on datasets with high-cardinality categorical attributes. Among optimization-based models, FIGS and Quant-BnB excels on numerical data but suffers larger train–test gaps and cannot exploit multi-way categorical splits. BONSAI bridges these extremes: its slot–node formulation, coupled with two search-space-reduction techniques, uncovers a compact set of highly informative multi-branch splits, preserving interpretability (fixed depth = 3) while matching—or surpassing—the predictive performance of state-of-the-art optmization-based trees.

**Interpretability and Practical Implications**  Visualizations of single DTs produced by C4.5, CART, and Quant–BnB are provided in the Appendix A.3. As illustrated in Figure 3 and Appendix A.3, BONSAI offers the best accuracy–complexity trade-off for practitioners seeking shallow, interpretable trees on mixed-type and imbalanced datasets. BONSAI achieves this by enabling multi-way splits on both numerical (highlighted in blue) and categorical (highlighted in yellow) features within a fixed depth, effectively compressing information. In contrast to one-hot encoded trees shown in Appendix A.3, BONSAI produces more compact and interpretable trees without compromising predictive performance.

**Limitations**  BONSAI is outperformed on four datasets: Zoo and Tic-tac-toe — both purely categorical with many low-arity attributes — where FIGS/Quant-BnB exploit exhaustive search over categorical splits; Car Evaluation — an extreme class-imbalance task (18.62 : 1) on which FIGS' class-weighted impurity is advantageous; Image Segmentation — a high-dimensional numerical dataset where FIGS again benefits from unrestricted depth during training.

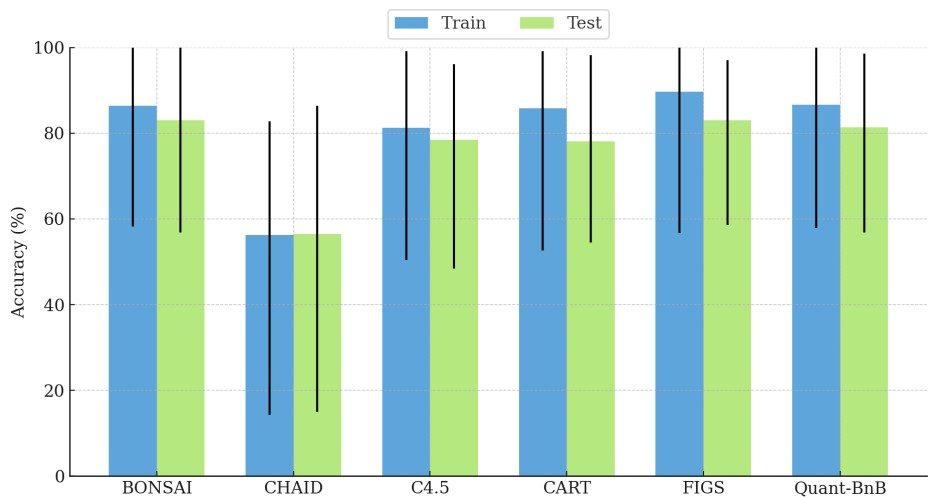

Figure 2: Average training and test accuracy across 17 datasets for each algorithm, with min-max error bars.

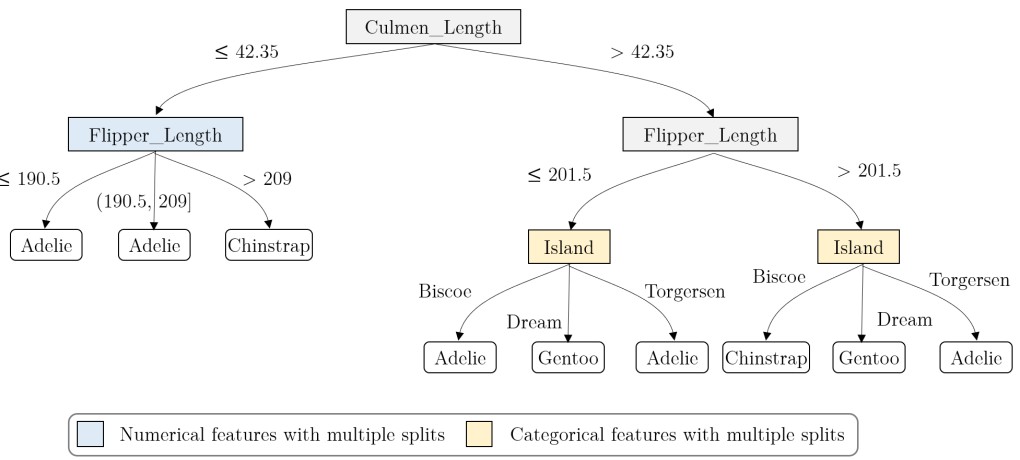

Figure 3: DT generated by BONSAI on the PENGUIN dataset, which contains mixed feature types and three output classes: `Adelie`, `Gentoo`, and `Chinstrap`.

**Future Work**    In this work, we employed fixed hyperparameters—such as sampling ratio and distance tolerance—uniformly across all features. This approach may overlook important properties of numerical features, including the number of unique values and inter-feature interactions. For categorical features, interpretability can be improved through post-processing by merging branches at leaf nodes that yield identical predictions (e.g., `Island = {Biscoe, Torgersen}`) into a single label. Future directions include: (1) adapting sampling ratio and distance tolerance based on feature characteristics, (2) post-pruning to enhance interpretability, and (3) incorporating additional objectives, such as precision and recall, to improve performance on highly imbalanced datasets.

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

# A  Technical Appendices

## A.1  Computational Complexity for Slot-Node Structure

In a binary DT where each $x_i$ contains continuous and/or binary features, the cost of computing an optimal tree of depth $d$ via exhaustive search is $\mathcal{O}(np^d)$ operations when all features are binary, and $\mathcal{O}(n^d p^d)$ operations when all features are continuous Mazumder et al. (2022) where $n$ denotes the number of unique feature values, $p$ the number of features. In contrast, for a non-binary and imbalanced DT constructed using the proposed slot-node structure, the worst-case cost of exhaustive search is

$$\mathcal{O}\left(\left(|F| \cdot 2^{u_{\max}}\right)^{s_{\max}}\right)$$

when all features are continuous ($F = F_{\text{num}}$), assuming the dataset remains undivided after each split. This complexity follows from the expression:

$$\prod_{s \in \{1, \ldots, s_{\max}\}} \sum_{f \in F} \sum_{b \in \{2, \ldots, b_{\max}\}} \binom{u(f, D)}{b},$$

where

$$u_{\max} = \max_{f \in F} u(f, D)$$

denotes the maximum number of unique values for any feature $f \in F$.

## A.2  Benchmark datasets

Table 4 lists the 18 classification benchmarks used in our study. For each dataset we report: (*i*) the total number of instances; (*ii*) the number of categorical features (**#Cat.**); (*iii*) the number of numerical features (**#Num.**); and (*iv*) the class–imbalance ratio, defined as the count of majority-class samples divided by minority-class samples. The corpus spans small (Zoo, 101 samples) to medium-sized (Rice, 3 810 samples) problems, with feature sets that are purely categorical (Tic-tac-toe), purely numerical (Wine), or mixed (ILPD). Imbalance levels range from perfectly balanced (Iris, Image Segmentation) to highly skewed (Car Evaluation, imbalance = 18.6), providing a diverse test bed for algorithms that must handle heterogeneous feature types and varying class distributions.

Table 4: Benchmark datasets used in our experiments. "#Cat." and "#Num." denote the number of categorical and numerical features, respectively. The imbalance ratio is the number of majority-class samples divided by the minority-class samples.

| Name | Abbreviation | #Instances | #Cat. Features | #Num. Features | Imbalance Ratio | #Classes |
|---|---|---|---|---|---|---|
| Zoo | Zoo | 101 | 16 | 0 | 10.25 | 7 |
| Iris | Iris | 150 | 0 | 4 | 1.00 | 3 |
| Hepatitis | Hepatitis | 155 | 13 | 6 | 5.15 | 2 |
| Wine | Wine | 178 | 0 | 13 | 1.48 | 3 |
| Penguin | Penguin | 344 | 2 | 4 | 2.15 | 3 |
| Wholesale | Wholesale | 440 | 1 | 6 | 6.72 | 2 |
| Indian Liver Patient Dataset | ILPD | 583 | 1 | 9 | 2.49 | 2 |
| Credit Approval | Credit | 690 | 9 | 6 | 1.21 | 2 |
| Blood Transfusion | Blood | 748 | 0 | 4 | 3.20 | 2 |
| Diabetes | Diabetes | 768 | 0 | 8 | 1.87 | 2 |
| Tic-tac-toe | Tic-tac-toe | 958 | 9 | 0 | 1.89 | 2 |
| Statlog (German Credit Data) | Statlog | 1000 | 13 | 7 | 2.33 | 2 |
| Banknote Authentication | Banknote | 1372 | 0 | 4 | 1.25 | 2 |
| Contraceptive | Contraceptive | 1473 | 7 | 2 | 1.89 | 3 |
| Car Evaluation | Car | 1728 | 6 | 0 | 18.62 | 4 |
| Image Segmentation | Image | 2310 | 0 | 19 | 1.00 | 7 |
| Rice | Rice | 3810 | 0 | 7 | 1.34 | 2 |

## A.3 VISUALIZATION OF DTs

Figures 4, 5, and 6 depict single DTs produced by Quant-BnB, CART, and C4.5, respectively. Unlike BONSAI, these trees rely exclusively on binary splits—either on numerical features or on one-hot encoded categorical features (e.g., `Island = Dream`). Notably, the tree generated by Quant-BnB is structurally balanced, with each internal node having both left and right children. FIGS is excluded from visualization due to its limited interpretability.

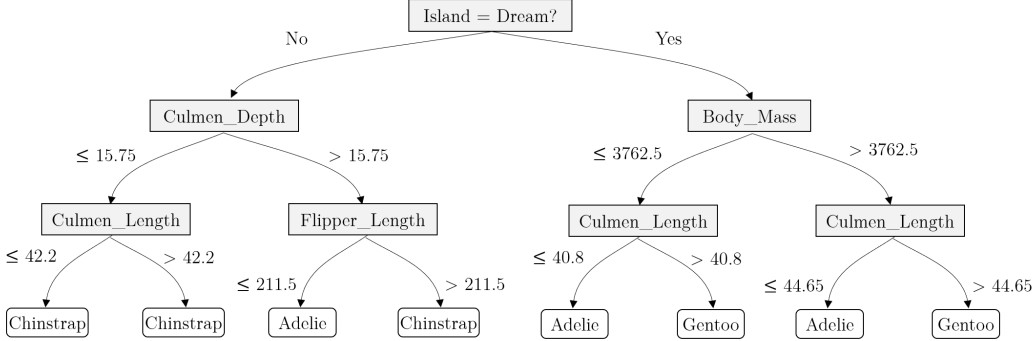

Figure 4: Illustration of DT from QuantBnB

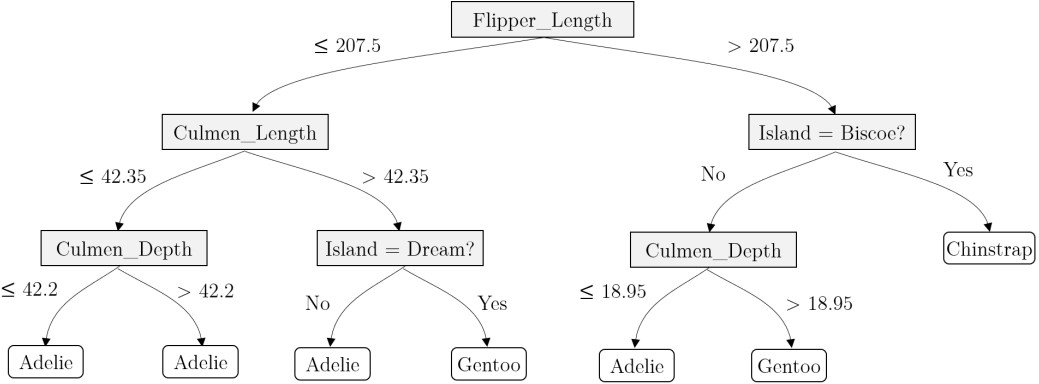

Figure 5: Illustration of DT from CART

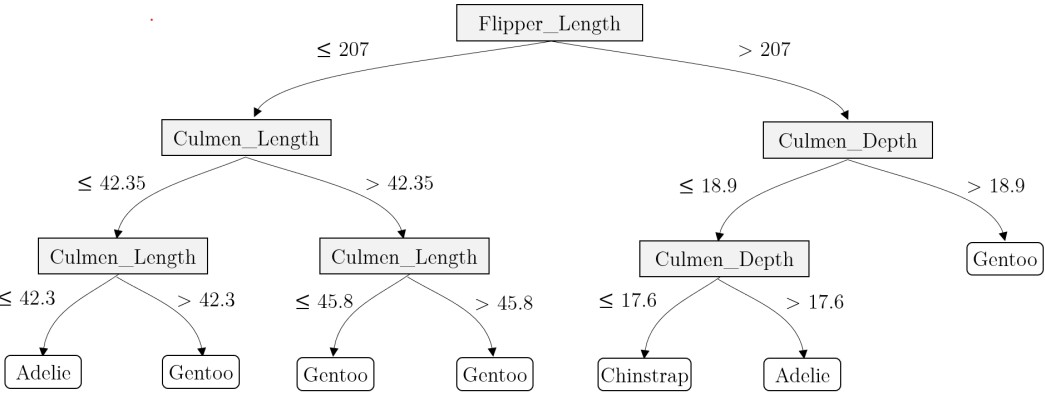

Figure 6: Illustration of DT from C4.5

