# OpenReview forum: "BONSAI: Depth-Constrained Decision Tree Induction via Approximation Optimization"
_ICLR.cc/2026/Conference — ICLR 2026 Conference Withdrawn Submission_

### Official Review · Reviewer_ybxf · 2025-10-29

**Soundness:** 2
**Presentation:** 1
**Contribution:** 1
**Rating:** 0
**Confidence:** 5

**Summary:**

The paper proposes BONSAI a new algorithm for constructing interpretable decision trees, where accuracy is the primary objective, and interpretation complexity is the second objective, while enforcing a tree depth constraint. The main idea of the method is to reduce the search space of the splits, by using certain rules, and then perform mixed-integer programming to optimize the objective function. The proposed method is evaluated seventeen public classification datasets and compared against reasonable baselines.

**Strengths:**

S1. The problem of constructing interpretable decision trees is well motivated and relevant to the ICLR conference.
S2. Some of the proposed ideas seem to have practical relevance.
S3. Extensive experimental evaluation using several real-world datasets.

**Weaknesses:**

W1. Many of the concepts used in the paper are not properly defined. For example the "slot-node structure", which is presumably a fundamental concept, is not properly defined and the reader is referred to the figure.
W2. While the proposed ideas and methods are reasonable, they are quite ad hoc. There is no proper justification and no theoretical analysis of the resulting algorithm.
W3. I would have preferred to see the presentation of the methods in text, rather than pseudocode. In my opinion, pseudocode should be used to provide additional information and clarify ambiguouities, it cannot be the primary source of information.
W4. I find the use of lexicographic optimization (which is not defined, btw), not very appropriate, since, if all highest-accuracy solutions have large complexity, then the complexity objective is not considered.
W5. Although the method is presented with focus on scalability, there are not results on running times.
W6. It is not clear that the improvement of BONSAI over FIGS and Quant-BnB is statistically significant.

**Questions:**

Q1. Can you provide any sort of theoretical analysis for your method, either about running time, or with respect to (approximate) optimality?

---

### Official Review · Reviewer_emg4 · 2025-10-30

**Soundness:** 2
**Presentation:** 2
**Contribution:** 2
**Rating:** 2
**Confidence:** 4

**Summary:**

The paper presents BONSAI, a novel algorithm for computing depth-constrained decision trees via approximate optimization. Specifically, the paper considers multiway branching based slot-based structure. To reduce the cost of computing optimal trees, BONSAI considers search-space reduction via candidate selection which allows the MIP model to be solved to optimality (however the result is no longer optimal).

**Strengths:**

Strengths:
------
- The paper presents a novel approach for constructing multiway decision trees
- The experimental results show the proposed approach outperforms the baselines in terms of test accuracy

**Weaknesses:**

Weaknesses:
------
Methodology:
- While the proposed approach is framed and motivated from an optimization perspective that is meant to overcome limitation of greedy approaches, the proposed methodology is not globally optimal, does not provide any theoretical guarantees. Perhaps more important, it does not actually lead to more “optimal” trees as evident by the lower *training accuracy* compared to Quant-BnB and FIGS. Further, while the search-space reduction technique is presented as a way of speeding up combinatorial optimization, there is no analysis of trade-off between runtime and performance as we control the parameters that reduce the state-space.
- As such, the success of the proposed methodology seems to be more related to the regularization provided by the minimal threshold distance and the sampling.
- The proposed methodology has limited technical novelty: minimal threshold distances have been considered before as noted in the paper (including in optimal decision trees, e.g., in [8]) and threshold sampling was also used as noted in the paper. Further, similar techniques could easily be encoded into other existing optimal decision tree approaches and potentially provide useful regularization.

Experiments:
- I am not convinced the comparison between the models is fair: there seems to be a uniform depth constraint across all models of maximum depth of 3. However, given that the proposed approach utilize multiple branches while approaches like Quant-BnB are utilizing binary splits, the trees generated by the proposed approach can be significantly more complex. While comparing binary trees under a maximum depth makes sense, when considering multiway trees, I am concerned this does not lead to fair comparison and perhaps a comparing under maximum number of splits is more appropriate?
- The paper misses many relevant baselines, especially considering the emphasis on multiple splits and regularization. These include stronger optimal baselines such as DPDT [1], Blossom [2], Top-K trees [3], TAO [4], modern multiple split approaches like Branches [5], and approaches that optimize complexity such as GOSDT [6] and SPLIT [7].
- Given the emphasis on regularization, it is not clear: (1) which hyper parameters were tuned for each baseline and how were they selected? (2) whether simple techniques like cost complexity pruning were used to improve generalization of techniques like Quant-BnB.
- No ablation and sensitivity analysis for the hyper-parameters of the proposed approach are provided.
- No runtime analysis and comparison
- No details on implementation (e.g., what solver is used) and the paper notes that one hyper parameter setting was used but it is not provided. In addition to the fact that the source code was not provided, this hurts the reproducibility of the results.

Also, the positioning of the work in the literature in Table 1 misses key relevant works: in particular, GOSDT (and SPLIT, and others) have considered complexity in their objective, and BRANCHES whose objective includes both accuracy and complexity and utilized multiple splits, and I believe it is not restricted to balanced trees.

[1] Kohler, H., Akrour, R., & Preux, P. (2025, August). Breiman meets bellman: Non-greedy decision trees with mdps. In Proceedings of the 31st ACM SIGKDD Conference on Knowledge Discovery and Data Mining V. 2 (pp. 1207-1218).

[2] Demirović, E., Hebrard, E., & Jean, L. (2023, July). Blossom: an anytime algorithm for computing optimal decision trees. In International Conference on Machine Learning (pp. 7533-7562). PMLR.

[3] Blanc, G., Lange, J., Pabbaraju, C., Sullivan, C., Tan, L. Y., & Tiwari, M. (2023). Harnessing the power of choices in decision tree learning. Advances in Neural Information Processing Systems, 36, 80220-80232.

[4] Carreira-Perpinán, M. A., & Tavallali, P. (2018). Alternating optimization of decision trees, with application to learning sparse oblique trees. Advances in neural information processing systems, 31.

[5] Chaouki, A., Read, J., & Bifet, A. Branches: Efficiently Seeking Optimal Sparse Decision Trees via AO. In Forty-second International Conference on Machine Learning.

[6] Lin, J., Zhong, C., Hu, D., Rudin, C., & Seltzer, M. (2020, November). Generalized and scalable optimal sparse decision trees. In International conference on machine learning (pp. 6150-6160). PMLR.

[7] Babbar, V., McTavish, H., Rudin, C., & Seltzer, M. (2025). Near Optimal Decision Trees in a SPLIT Second. arXiv preprint arXiv:2502.15988.

[8] Shati, P., Cohen, E., & McIlraith, S. A. (2023). SAT-based optimal classification trees for non-binary data. Constraints, 28(2), 166-202.

**Questions:**

I would appreciate the authors' response to the points listed under weaknesses above

---

### Official Review · Reviewer_5hAB · 2025-11-02

**Soundness:** 2
**Presentation:** 2
**Contribution:** 3
**Rating:** 2
**Confidence:** 4

**Summary:**

The paper introduces BONSAI, a new algorithmic framework for decision tree induction under depth constraints, formulating the problem as an approximation optimization task rather than a full mixed-integer program (MIP).

**Strengths:**

1. BONSAI provides a clear mathematical formulation integrating lexicographic optimization over accuracy and interpretation complexity.
2. The proposed slot–node structure and two-stage candidate selection (threshold distance control and random sampling) effectively reduce the search space while maintaining optimality within the reduced domain
3. The formulation directly supports multi-way splits on both numerical and categorical attributes without relying on one-hot or itemset encodings.

**Weaknesses:**

1. The paper does not report any computational runtime, solver statistics, or scalability experiments, even though the core claim revolves around reducing MIP complexity through the slot-node approximation.
2. The comparisons are based solely on point estimates of test accuracy, without reporting variability (mean ± std) or statistical significance across multiple runs. Moreover, comparing a non-optimal approximate method against globally optimal solvers only on test accuracy can be misleading; a fairer baseline would also analyze training accuracy vs. optimal training objective value to understand how far BONSAI deviates from optimality.
3. It also appears that BONSAI employs multi-way splits, while most baselines (C4.5, CART, Quant-BnB, FIGS) are restricted to binary splits. This structural asymmetry makes the comparison potentially unfair, BONSAI has greater expressive capacity under the same depth constraint. The authors should justify whether the “optimal” trees from other models are re-optimized under comparable multi-way configurations.
4. All datasets used are tiny (<= 4k samples) and low-dimensional, yet the paper claims efficiency and generality. With such small instances, even baseline MIP solvers can already reach the global optimum. Thus, BONSAI’s advantages on larger or high-dimensional data remain unverified.
5. The paper omits relevant references in global optimization for decision trees, most notably "A Scalable Deterministic Global Optimization Algorithm for Training Optimal Decision Trees" (Kaixun Hua et al.), which directly addresses scalability and deterministic search.
6. The paper does not specify which MIP solver or parameter settings were used (e.g., Gurobi, CPLEX, time limits, tolerance levels).

Overall, While BONSAI presents a new problem formulation (slot-node structure and approximation optimization), the paper fails to substantiate its computational or theoretical advantages. Key omissions: runtime data, variance analysis, fairness of baselines, and missing related work prevent the reader from assessing whether the method genuinely advances the state of optimization-based decision tree learning.

**Questions:**

Refer to the weaknesses.

---

### Official Review · Reviewer_eE8k · 2025-11-04

**Soundness:** 2
**Presentation:** 2
**Contribution:** 2
**Rating:** 2
**Confidence:** 3

**Summary:**

The paper presents BONSAI, a depth-constrained decision-tree induction framework. It first enumerates candidate splits within a slot–node scaffold using (i) a minimum-distance filter and (ii) sampling (𝜏,𝜌), then solves a mixed-integer program (MIP) over that pruned set with a lexicographic objective (minimize error, then interpretation complexity). Experiments compare against standard tree learners on mostly small tabular datasets under a fixed maximum depth (𝑑=3).

**Strengths:**

- The writing of this paper is easy to follow.
- This paper is studying a very important research problem.

**Weaknesses:**

-  (1) the algorithmic superiority of BONSAI is not justified theoretically or empirically;
- (2) Algorithm 1 and Algorithm 2 lack time/space complexity analyses, and the paper does not report runtime/memory/solver statistics or head-to-head training-time comparisons;
- (3) there is no comparison to state-of-the-art scalable optimal tree methods, and most datasets are small—leaving scalability unsubstantiated;
- (4) the evaluation protocol is unfair: fixing depth (d=3) creates a capacity mismatch that favors methods able to activate more branches per internal node; fair comparisons should control model complexity and use cross-validation with matched tuning.


My main comments are as follows:

- BONSAI is motivated as “approximation optimization”—search a reduced candidate space, then solve an exact MIP. However, the paper does not establish conditions under which this pipeline is provably better (accuracy, or interpretability) than strong baselines, nor does it characterize the gap induced by candidate pruning (τ,ρ) relative to the unrestricted objective. Without theory (e.g., bounds, guarantees) or decisive empirical evidence (large-scale, time/space reporting), the claimed advantages remain assertions.

- The core contribution hinges on Algorithm 1 (slot-wise orchestration) and Algorithm 2 (candidate generation/SEARCH_NODE). Yet the paper provides no big-O bounds (time and memory) as functions of (n,p,d,p_max,τ,ρ), nor worst-case vs. practical scaling. Because BONSAI culminates in a MIP whose size depends directly on the number of retained candidates, omitting these analyses makes feasibility and scalability impossible to judge.

-  Experiments are primarily on small datasets, with no wall-clock training time. There is also no comparison against strong scalable optimal tree models. The proposed method BONSAI actually depends on the multi-way splits strategy, such strategy has been applied in optimal classification trees. For example, Subramanian, S., & Sun, W. (AAAI 2023), Scalable optimal multiway-split decision trees with constraints, which demonstrates strong performance at scale on multiway splits. The absence of such head-to-head comparisons weakens the significance of the contribution. Consequently, the paper does not demonstrate that BONSAI is either better or faster where it matters.

- Fixing depth (d=3) biases comparisons when methods differ in effective capacity per node. The multi-way decision trees are more likely to achieve performance at shallow depths because they can generate more decision rules. For a fair test, control capacity (equal number of leaves/nodes or an explicit complexity budget/penalty), and use stratified k-fold cross-validation with matched hyperparameter tuning and statistical tests (CIs or paired tests). Without this, reported accuracy improvements can simply reflect larger capacity, not a better learning principle.

**Questions:**

See the weaknesses.

---

### Note · Authors · 2026-03-17

I have read and agree with the venue's withdrawal policy on behalf of myself and my co-authors.

---

### Meta-Review · Area_Chair_pZ9K · 2026-01-07

**Summary:**

The paper proposes BONSAI, a depth-constrained decision tree method that prunes candidate splits and then solves a reduced MIP with a lexicographic objective. Reviewers agree the problem is relevant and the idea is reasonable. However, they find the main claims insufficiently supported. Key issues include missing runtime and scalability evidence, unfair experimental comparisons due to multi-way splits under fixed depth, lack of statistical validation, missing strong baselines, and limited clarity and reproducibility. Overall reviewer sentiment is clearly negative.

**Reviewer Concerns:**

Addressed or partially addressed:
- Some clarification was provided on the general motivation and formulation, but this does not resolve the core concerns.

Still outstanding:
- No runtime, memory, or solver statistics despite claims about efficiency.
- Scalability is not demonstrated.
- Comparisons are unfair due to multi-way splits versus binary baselines under fixed depth.
- No control for model capacity, no cross-validated tuning, and no significance testing.
- Missing ablations on pruning and sampling parameters.
- Important related work is omitted or insufficiently discussed.
- Reproducibility is weak due to missing solver details and implementation information.

**Reviewer Scores:**

I do not expect meaningful score changes after discussion. All reviewers raising rejection recommend maintaining their scores.

---

### Decision · Program_Chairs · 2026-01-26

Reject